# Uncovering the endogenous features of potassium salts' global transfer: A complex network perspective

Bo Zhang, Wensong Zhang 🄳 *

School of Economics and Management, Beijing Jiaotong University, Beijing, China

* qq300120@126.com

## Abstract

Potassium is a decisive strategic resource to ensure food safety production and supply, which many nations define as a critical metal. Due to the unbalanced distribution of resources and production capacity and the separation of the primary potassium-consuming and supplying countries, international trade is the main supply channel for potassium-consuming countries to acquire enough resources. Understanding the characteristics of potassium trade networks and the evolution of trade patterns is essential for supply security. To explore this issue, this paper employs the complex network theory to quantitatively analyze the evolution characteristics of the global potassium trade network (PTN) from 2000 to 2021. The results show (1) Overall, the potassium trade shows a trend of gradual prosperity, efficiency, and concentration; (2) During the two decades, the main exporting countries of potassium remained stable, while imports changed significantly; (3) The evolution of the potassium trade community has characterized the fragmentation-regionalization-high concentration over time; (4) The trade flow of PTN is unbalanced, and few countries show outstanding capabilities but a single function. These findings would help trade policymakers manage the supply of strategic raw materials more effectively.

## 1. Introduction

Potassium is mainly used in the production of potash fertilizers, which are widely used in agriculture and is a critical strategic resource to ensure food safety production and supply. Due to the sharp increase in demand and the scarcity of resources, potassium has been defined as a critical metal by many countries such as China, the United States, Thailand, and Brazil [1–4]. However, the supply of potassium is mainly concentrated in North America, Belarus, and Russia, while consumption is concentrated in countries such as China, Brazil, and India [5]. There are deviations between suppliers and consuming countries of potassium, and trade is the main global potassium salt supply channel. The trading form of potassium in the international market is mainly primary processed products. Hence, the potassium trade can be replaced by the potassium salt trade [6]. The rapid growth in demand for potassium salt and the excessive concentration of resources have enhanced competition among countries, threatening the stability

**Funding:** The funders had no role in study design, data collection and analysis, decision to publish, or preparation of the manuscript.

**Competing interests:** The authors have declared that no competing interests exist.

of the potassium trade. Maintaining the stability of the potassium trade is essential to food safety production and national economic development.

Global potassium salt production is concentrated, and Canada, Russia, and Belarus account for 63% of total production in 2020 [7]. The international potassium supply is in a monopoly pattern, which is not conducive to the security of the global potassium supply [8]. Due to the uncertainty caused by the COVID-19 pandemic, most of the world's economies had to follow the measures taken by their governments, which is likely to cause disruptions in supply chains [9]. Russia is a significant producer and exporter of potassium salt. With the outbreak of the Russia-Ukraine conflict and the increase of sanctions against Russia, many countries have begun to restrict and review the goods related to Russia [10]. Under the current potassium supply trade system, enormous demand pressure and fragility factors such as geopolitics, war, and the COVID-19 epidemic have increased the risk of interruption of the potassium supply chain [11]. The instability of trade will affect the supply of resources and damage the economy of trading countries. Therefore, studying the evolution of the potassium trade pattern is of great significance to the development of the global economy.

However, as far as we know, although some scholars have begun to pay attention to the trade of potassium, it has yet to receive enough attention. Wang and Kong constructed a complex network model of potassium trade to study the evolution characteristics of potassium country trade, focusing on trade scale, trade relationship, and trade flow distribution [12]. Song et al. employed the material flow analysis to establish the framework of the life cycle of potassium salt in China, revealing the spatiotemporal characteristics of the main potassium flow [13]. Existing studies have made some efforts in the potassium salt trade. Nevertheless, there is no systematic analysis of the trade pattern from a global perspective, and the evolution characteristics of the potassium salt international trade pattern still need to be completed.

Trade networks are complex dynamic systems with complex trade relationships between countries [14]. Traditional trade models cannot explain the highly heterogeneous characteristics of international trade networks [15]. Academically, the complex network can reveal many new features and topology dynamics of international trade entirely and partially [16]. Complex network theory is currently widely used to analyze the topological characteristics, time evolution characteristics, community evolution, and competition pattern of mineral resource trade networks, such as coal [17], fossil fuel [18], boron ore [19], lithium [20], nickel [21], barite [22]. The evolution of the global potassium salt trade pattern still needs further study. It is worth noting that studying the international trade pattern of potassium salt is essential because of the research gap in the academic field and global geopolitical instability, such as the recent Russia-Ukraine conflict that may affect the global potassium salt supply. Exploring the evolution of the international potassium salt trade pattern and identifying the critical countries in PTN is crucial for trading countries to formulate trade strategies and ensure the supply security of potassium salt.

In summary, although previous studies on the evolution characteristics of PTN have been studied, these still need to be improved. Firstly, the research is limited to a simple analysis of a specific country and needs an analysis of the evolution of the global trade pattern. Secondly, the core countries in PTN have yet to be identified. To fill this gap, this paper selects potassium salt trade data from 2000 to 2021 to construct a complex network model of global potassium salt, which is used to explore the evolution characteristics of trade patterns and identify the role of different countries in the network. This paper has two contributions: (1) the study is the first to focus on the evolving characteristics of the global trade pattern of potassium salt, providing support for government policymakers to formulate potassium trade policies in different economies. (2) This study identifies the major economies and their roles in international trade in potassium salt. This study expects to provide decision support for building a globally

stable and efficient potassium salt supply by analyzing the evolution characteristics of the global potassium salt trade pattern.

This paper is organized as follows: Section 2 is the data and method. Section 3 presents the results and our main findings. Section 4 is conclusions and policy implications.

## 2. Methodology

### 2.1 Data

This paper selects the import trade data of potassium salt from 2000 to 2021, which HS code is 310420, and the description is fertilizers, mineral or chemical; potassic, potassium chloride. All data selected for this study are from UN Comtrade. This study selects the import data of potassium salt instead of export data due to the asymmetry in bilateral merchandise trade [23]. We clean and preprocess the original data and eliminate some invalid statistical units and trade relations with zero trade volume.

### 2.2 Complex network construction

The complex network model of potassium salt international trade C = (V,E), where V is the set of nodes, and E is the set of edges between nodes. V = {$v_i$: $i$ = 1, 2, . . ., $n$}, and n is the number of nodes. E = {$e_i$: $i$ = 1, 2, . . ., $m$}, and m is the number of edges. The matrix of the complex network model is:

$$C = (V, E) = \begin{bmatrix} 0 & w_{1,2} & \cdots & w_{1,n} \\ w_{2,1} & 0 & \cdots & w_{2,n} \\ \vdots & \vdots & \ddots & \vdots \\ w_{n,1} & w_{n,2} & \cdots & 0 \end{bmatrix} \tag{1}$$

In PTN, the nodes are countries, the edges are trade relations, the direction of the edges is the flow direction of potassium salt, and the weights of the edges are the trade volume.

### 2.3 Topological properties

**2.3.1 Degree and degree distribution.** The degree ki of a node vi in a graph is the number of edges linked. It is the sum of the in-degree $k_i^{in}$ and the out-degree $k_i^{out}$, which represent the number of import and export relationships of a country. The greater the degree of a node, the more significant the node's role in the network. As shown in Eq (2),

$$k_i = k_i^{in} + k_i^{out} = \sum\nolimits_{j=1}^{n} d_{ji} + \sum\nolimits_{j=1}^{n} d_{ij} \tag{2}$$

Where $i$ and $j$ are nodes in the network, and $d$ refers to the actual trade relationship between country $i$ and country $j$.

The degree distribution describes the distribution characteristic of the node's number of connections [24]. If the degree distribution of all nodes in a network follows Eq (3), then the network owes scale-free characteristics, which means a small number of nodes have many connected edges.

$$P(k) \propto k^{-\gamma} \tag{3}$$

Where $P(k)$ is the proportion of nodes with degree $k$ in the entire network, which is used to describe the distribution function of the network. $\gamma$ is the power-law exponent.

**2.3.2 Strength.** The degree can only represent the trade relationship of nodes but not the strength of the relationship between trade nodes. Hence, this paper introduces strength, the total weight of all edges connected to the node. Strength includes in-strength $S_i^{in}$ and out-strength $S_i^{out}$ which represent a country's imports and export, respectively. It is defined as Eqs (4) and (5):

$$S_i^{in} = \sum_{j=1}^{N} w_{ji} \tag{4}$$

$$S_i^{out} = \sum_{j=1}^{N} w_{ji} \tag{5}$$

where $w_{ji}$ and $w_{ji}$ represent the edge $(j, i)$ and $(i, j)$ weights, respectively.

**2.3.3 Node centrality.** Betweenness centrality measures the intermediary ability of the nodes as mediums in the network. It reflects the node's ability to control the flow of resources in the network. The greater the value, the stronger the intermediary ability of the node. In the international trade network of potassium salt, betweenness centrality is the frequency at which a country stands on the shortest path between two other countries. The betweenness centrality of node $i$ is as follows [25]:

$$BC_i = \sum_{x \neq i \neq y} \frac{\sigma_{xy}(i)}{\sigma_{xy}} \tag{6}$$

Where $\sigma_{xy}$ is the total number of shortest paths from node $x$ to node $y$, and $\frac{\sigma_{xy}(i)}{\sigma_{xy}}$ is the number of these paths that pass node $i$.

The closeness centrality reflects what degree a country stands in the central position of the network. The higher the closeness centrality, the closer the economy is to other economies, and this indicator reflects an economy's ability to anticontrol and autonomy. The more central a country is, the lower its total distance from all other nodes. The closeness centrality of node $i$ is as follows [26]:

$$CC_i = \frac{1}{\sum_{i \neq t} d(i,j)} \tag{7}$$

Where $d(i, j)$ is the distance between node $i$ and node $j$, the minimum length of any path connecting node $i$ and node $j$. The length of a path is the sum of the weights of its edges.

**2.3.4 Community detection.** In international trade networks, some countries are relatively closely connected. They often form a community in which the essence is to cluster and merge closely connected similar nodes in the network. Analyzing the community structure of international trade networks can better understand the characteristics of trade patterns at the regional scale [27]. Modularity can reflect the degree of globalization of trade and the degree of community differentiation. Analyzing the choice of trade partners and the evolution of trade relations between different trade groups is currently the most commonly used method to measure the structural strength of network communities. It is defined as Eq (8) [28]:

$$M = \frac{1}{2m} \sum_{ij} \left( u_{ij} - \frac{k_i k_j}{2m} \right) \delta\left( c_i, c_j \right) \tag{8}$$

Where m is the total weight of all edges in the network, this paper represents the total global trade volume. $u_{ij}$ is the weight of the edge from node $i$ to node $j$. $k_i$ and $k_j$ is the sum of the weights of all edges connected to node $i$ and node $j$. $\delta(c_i, c_j)$ is the discriminant function. If two countries $i$ and $j$ are in the same community, then $\delta(c_i, c_j) = 1$, otherwise, $\delta(c_i, c_j) = 0$.

The larger the modularity, the clearer the community structure of the network. Modularity between 0.3 and 0.7 indicates that the network has an obvious group structure [29].

**2.3.5 Roles of major countries.** Degree and betweenness centrality are indicators commonly used to measure the importance of nodes in complex networks [30]. The node degree emphasizes the number of edges directly connected to the node, which can explain the node's importance to a certain extent. However, in a complex network, nodes with the same degree may play different roles [31]. Countries with large trade volumes are generally considered to play an essential role in trade networks. Nodes with high betweenness centrality usually play a vital role in maintaining effective connections between nodes in the network [32]. Hence, this study identifies a country's role in a trade network through trade relations, volume, and intermediary capacity.

# 3. Results

## 3.1 Basic topological properties

**3.1.1 Degree and degree distribution: A scale-free network.** As shown in Fig 1(a), the global potassium salt trade volume increased from 24.6Mt in 2000 to 57.2Mt in 2021, showing a rapid upward trend. Affected by the global financial crisis in 2008, the trade volume reached its lowest point in 2009. Subsequently, as the global economy recovered, potassium salt trade volumes gradually returned to pre-financial crisis levels. In 2016, due to the slump in the price of potassium salt, the trade volume fell to the bottom again. Overall, the global trade volume has experienced three distinct growth periods (2000–2008, 2010–2015, 2017–2021) and two low points (2009, 2016).

Scale-free features usually exist in many real networks, especially in trade networks [33]. To explore the distribution characteristics of trade relations among different countries, this paper draws the cumulative degree distribution scatter diagrams in 2000, 2005, 2010, 2015, and 2021 (Fig 1(b)). It examines the structure and degree distribution characteristics of PTN. The results show that PTN approximately follows the scale-free characteristic, which also means that the importance of nodes is highly heterogeneous. In other words, PTN is a scale-free network where minority countries have many trade relationships, and majority countries have few trade relationships. Minority economies with significant trade relationships are prominent in the network and are more likely to influence the stability of the global potassium salt trade.

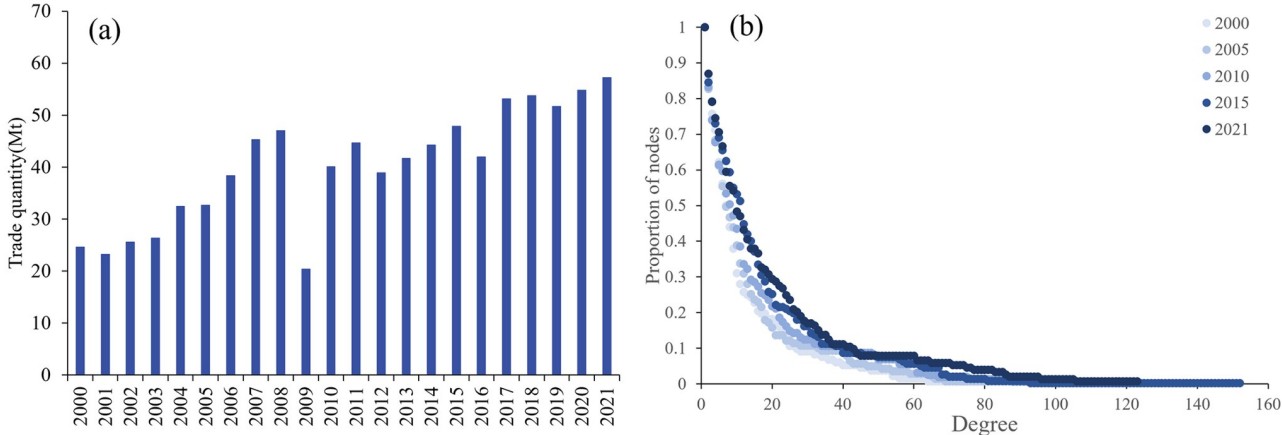

**Fig 1. Potassium salt trade quantity from 2000 to 2021(a) and cumulative distribution of PTN in 2000, 2005, 2010, 2015, and 2021(b).**

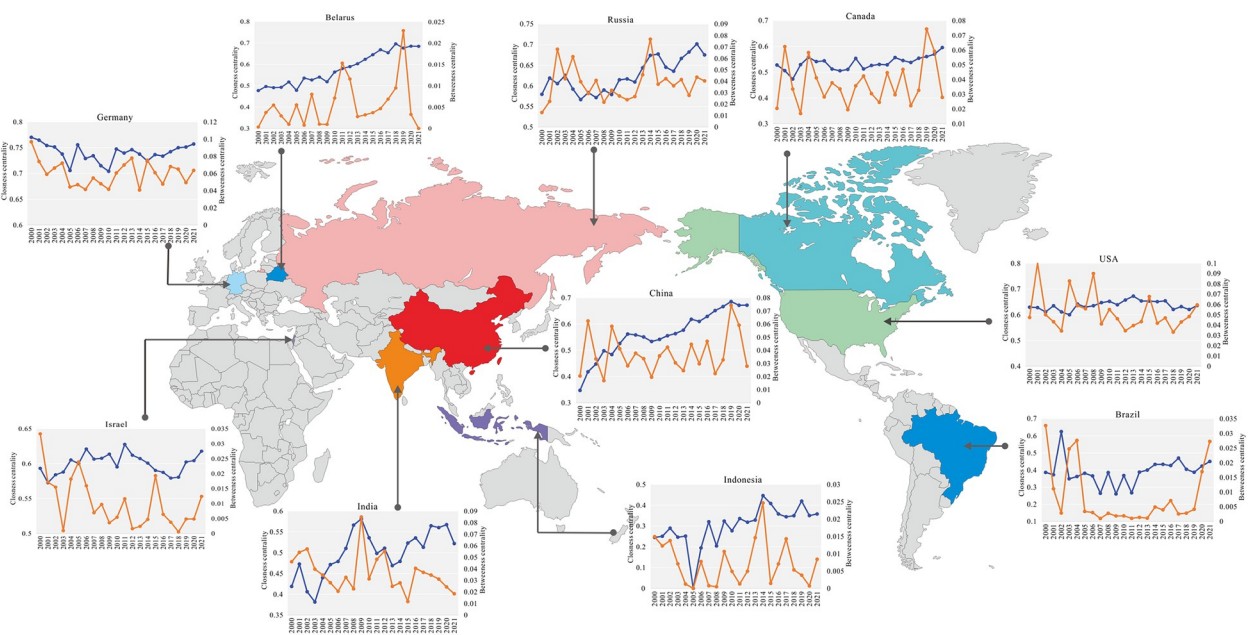

**Fig 2. Closeness centrality and betweenness centrality ranking changes of key countries in potassium salt trade from 2000 to 2021.**

**3.1.2 Nodal and network centrality.** This paper selects the countries with higher frequency in the top 10 of betweenness centrality and closeness centrality (Fig 2). Major potassium salt suppliers such as Russia, Belarus, Canada, and China have shown a steady rise and relatively high closeness centrality, indicating that suppliers have more discourse power in PTN. This phenomenon is consistent with the view of Yuxuan et al. (2022) that the international potassium salt supply is a monopoly pattern. In international trade, a network where a minority of trade relationships control the majority of trade volume is more likely to experience a supply crisis when exposed to external shocks [34]. Major potassium salt-consuming countries such as India, Indonesia, and Brazil have low closeness centrality and fluctuate continuously during the research period. This indicates that the anticontrol capabilities and the ability to obtain resources are low.

Regarding betweenness centrality, we find that developed countries such as Germany, France, and the United States have higher betweenness centrality (S1 Table), representing that developed countries in Europe and America also have a certain discourse power and a relatively high control power in PTN. It is worth noting that China, as an essential supplier and consumer country, is playing an increasingly important role in PTN, and the ranks of betweenness centrality and closeness centrality are both high and relatively stable. China is getting closer and closer to the central position, which shows that China maintains a high level of trade control power and is gradually improving its anticontrol capabilities, which the position and influence of PTN are increasing.

## 3.2 Evolution of trade patterns

**3.2.1 Trade flow characteristics.** In the two decades, the exporting countries of PTN have remained stable, mainly Russia, Canada, Germany, Israel, Belarus, and Jordan, while the importing countries have had significant changes. We map the global potassium salt trade flows in 2000, 2010, and 2021 and summarize how trade patterns have evolved (Fig 3). In

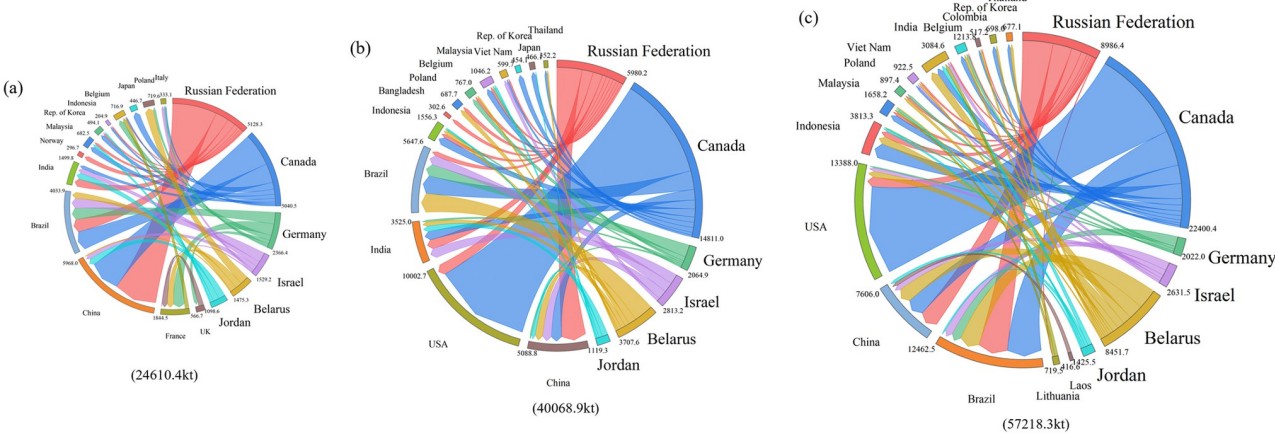

**Fig 3. Potassium salt trade of major countries in the world in 2000 (a), 2010 (b), and 2021 (c).**

2000, the major importing countries were China, Brazil, India, France, Etc. China's potassium salt mainly came from Russia (2960.3kt) and Canada (2198.2kt), accounting for 49% and 36% of China's total imports, respectively. In 2010, the United States became the world's largest potassium salt importer, accounting for 25% of the world's total imports. Canada became the United States' primary source of potassium salt, while China's potassium salt mainly came from Russia. The major importing countries in 2021 are the United States, Brazil, China, Indonesia, Etc. It is worth noting that Canada, as the largest exporter of potassium salt, mainly exports to the United States and Brazil, which means that the global potassium salt trade center will shift from Asia to America. China's imports slowly increase as the world's largest potassium salt consumer. The reason is that the Chinese government released the "National Mineral Resources Planning (2016–2020)" in 2015, in which the potassium salt bases in Qarhan, Qinghai, and Lop Nur, Xinjiang, were chosen as the national potassium reserve bases. This led to an increase in domestic potassium mining in 2016 and improved China's potassium self-sufficiency rate.

Notably, the discovery and development of potassium deposits in Canada have changed the global supply pattern. The UK's plans for large-scale polyhalite mining will provide an additional source of potassium, which may once again change the global supply pattern.

**3.2.2 Community structure.** The trading community is made up of interconnected trading economies. Nodes in the same community are closely connected, and these nodes usually have common attributes or similar functions. As per Eq (7), this paper quantifies the modularity of countries in the PTN and subsequently categorizes communities according to their modularity. The global potassium salt trade has prominent regional characteristics, forming 2–4 communities, and the rest are marginal trading countries. These small trading countries are separated because of their small trade volume and few surrounding trading countries. Fig 4 is the pattern of global potassium salt trade communities in 2000, 2010, and 2021. We choose the name of the country with the most significant trade volume in the community to name the corresponding community, such as Canada-community.

In 2000, global potassium salt trade flows were fragmented, mainly forming two communities: Canada-community and Russia-community. Canadian community mainly includes China, India, Brazil, Etc., and Russia-community mainly in developed countries in Europe and America, such as the United States and Germany. Different from the fragmentation of

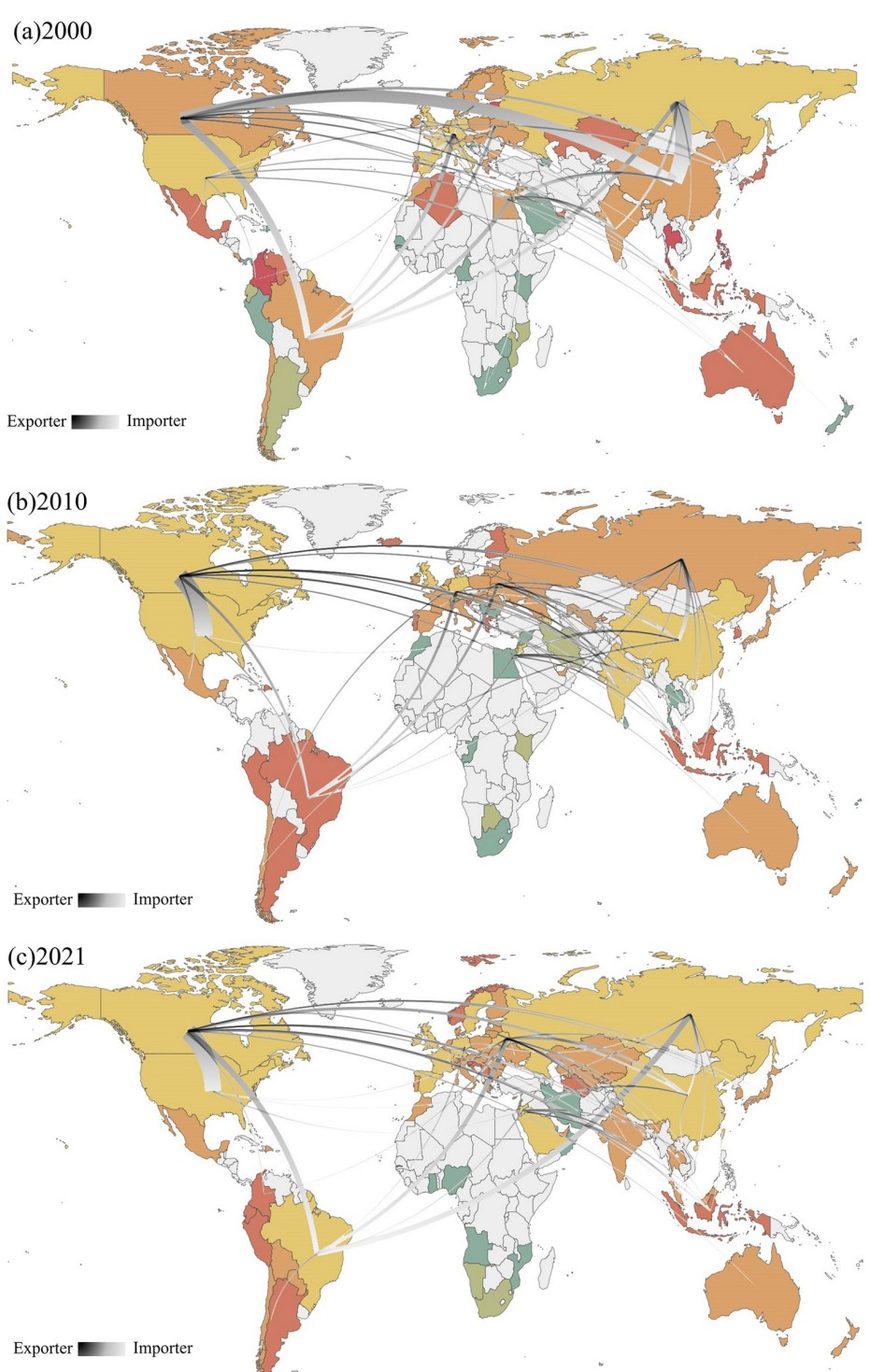

**Fig 4. Community evolution of potassium salt trade.** (Note: The same color indicates the same community. Lines are the trade flows of major countries).

communities in 2000, the evolution of communities in 2010 has apparent geographical and spatial distribution characteristics, forming three major communities: North America, South America, and Europe. The United States separated from the Russia-community and joined the Canada-community, which has become an essential importer of potassium salt in PTN. European countries, such as France and Germany, joined the Russia-community. In addition, Brazil, Peru, and Argentina form the South American community. This shows that the potassium salt trade is significantly regionalized, and different trade preferences in different regions form different trade communities. Compared to 2010, the evolution of the community in 2021 has a significant change. The countries in Russia-community have nearly joined the Canada-community. In other words, the world's major potassium salt suppliers and consumers have gradually become a trading community. With international trade globalization, the potassium salt trading community is also converging, gradually forming an efficient, stable, but centralized system. Moreover, it is worth noting that geopolitically China is more likely to be in the same community as Russia. However, China and Canada have always been in the same community.

During the research period, the global potassium salt trading community experienced the development of fragmentation-regionalization-high concentration. Trade preferences have shifted across different regions. The top exporting country is the core country in the community, and different consuming countries join different trading communities according to their different trade preferences.

## 3.3 Roles of major countries

This paper identifies the core countries in the network based on their trade relations ($k_i^{in}$ and $k_i^{out}$), trade volumes ($S_i^{in}$ and $S_i^{out}$), and intermediary capabilities ($BC_i$), which together determine the country's role. Where $k_i^{in}$ and $S_i^{in}$ indicate a country's import capacity, $k_i^{out}$ and $S_i^{out}$ indicate a country's export capacity, $BC_i$ indicates a country's ability to control resources in the network. Table 1 shows the top 5 countries of those indicators in 2000, 2010, and 2021, and it can find that the main trading countries in potassium trade are Canada, Russia, Belarus, Israel, Germany, France, the United States, and the United Kingdom, China, Brazil, India, Indonesia, and Malaysia. Fig 5 is the radar chart drawn by the five indicators normalized by the minimum-maximum method.

The polygons in the radar chart represent a country's trade capacity. The larger the area of the polygon in the radar chart, the stronger the comprehensive capability of the corresponding country. According to the evolution characteristics of each country's trade relations, trade volume, and intermediary capacity, this paper divides 13 countries into three categories. The first group is Russia, the United States, China, and France. These countries show certain import and export capabilities and relatively strong intermediary capabilities. It also represents that these four countries have sufficient key information and a stable and critical transit position in the potassium salt trade. Besides, China has a great demand for potassium and is a major importer of potassium salt, but the number of countries that export potassium salt to China is four times the number of importing countries. The second group consists of major suppliers such as Germany, Israel, and Belarus, which have been important exporters in recent years. The third group consists of countries such as Brazil, India, Malaysia, Canada, the United Kingdom, and Indonesia, which present a relatively single ability. We have classified Canada, the largest potassium salt exporter in the network, into the third category of countries. Although Canada contributes the most significant export volume globally, its trade structure is single, most of which flows into the United States. Canada displays high trade intensity yet weak trade relations and intermediary capabilities.

**Table 1. Top 5 countries by five different indicators in 2000, 2010, and 2021.**

| | Rank | 2000 | | 2010 | | 2021 | |
|---|---|---|---|---|---|---|---|
| | | Country | Value | Country | Value | Country | Value |
| In-degree ($k^{in}$) | 1 | Brazil | 19 | France | 23 | Netherlands | 36 |
| | 2 | Malaysia | 19 | Malaysia | 21 | France | 29 |
| | 3 | Indonesia | 19 | Thailand | 20 | Malaysia | 25 |
| | 4 | France | 17 | India | 19 | Spain | 25 |
| | 5 | Spain | 16 | Colombia | 18 | Germany | 23 |
| | Rank | 2000 | | 2010 | | 2021 | |
| | | Country | Value | Country | Value | Country | Value |
| Out-degree ($k^{out}$) | 1 | Germany | 88 | Germany | 90 | Germany | 92 |
| | 2 | USA | 53 | USA | 75 | Belarus | 76 |
| | 3 | Russian Federation | 50 | Russian Federation | 62 | Russian Federation | 75 |
| | 4 | United Kingdom | 45 | Israel | 53 | China | 71 |
| | 5 | Israel | 42 | Belgium | 52 | USA | 64 |
| | Rank | 2000 | | 2010 | | 2021 | |
| | | Country | Value | Country | Value | Country | Value |
| In-strength ($S^{in}$) | 1 | Brazil | 4.3 | USA | 10.1 | USA | 13.5 |
| | 2 | China | 3.0 | Brazil | 6.1 | Brazil | 12.8 |
| | 3 | India | 2.0 | China | 5.2 | China | 7.7 |
| | 4 | France | 1.7 | India | 4.4 | Indonesia | 4.0 |
| | 5 | Malaysia | 1.1 | Indonesia | 1.8 | India | 3.2 |
| | Rank | 2000 | | 2010 | | 2021 | |
| | | Country | Value | Country | Value | Country | Value |
| Out-strength ($S^{out}$) | 1 | Canada | 6.0 | Canada | 15.8 | Canada | 24.4 |
| | 2 | Russian Federation | 5.9 | Russian Federation | 7.1 | Russian Federation | 10.1 |
| | 3 | Germany | 3.3 | Belarus | 4.8 | Belarus | 9.8 |
| | 4 | Israel | 2.1 | Israel | 3.3 | Israel | 3.4 |
| | 5 | Belarus | 1.5 | Germany | 3.2 | Germany | 3.3 |
| | Rank | 2000 | | 2010 | | 2021 | |
| | | Country | Value | Country | Value | Country | Value |
| Betweenness centrality | 1 | Germany | 0.097 | France | 0.055 | Germany | 0.064 |
| | 2 | France | 0.078 | USA | 0.055 | USA | 0.060 |
| | 3 | USA | 0.047 | Germany | 0.042 | Netherlands | 0.055 |
| | 4 | United Kingdom | 0.047 | China | 0.036 | Russian Federation | 0.040 |
| | 5 | India | 0.046 | India | 0.031 | Spain | 0.036 |

By observing the radar chart, we found that only some countries have shown significant capabilities in PTN. In other words, exporting countries only show strong export capabilities, and importing countries only show strong import capabilities. This phenomenon indicates that the trade flow of the PTN is unbalanced. The more significant the difference in the distribution of trade flows between countries with less trade relations, the smaller the distribution of trade flows between countries with more trade relations. Moreover, the distribution of trade flows becomes increasingly different over time. From a community perspective, the potassium salt trade has evolved over time and towards forming trading groups. Countries choose different trading partners according to their trade preferences, but resource access depends on the core countries of their group. The current potassium salt supply-trade pattern presents monopoly characteristics, and an overly concentrated trade relationship poses hidden dangers to the long-term stable supply.

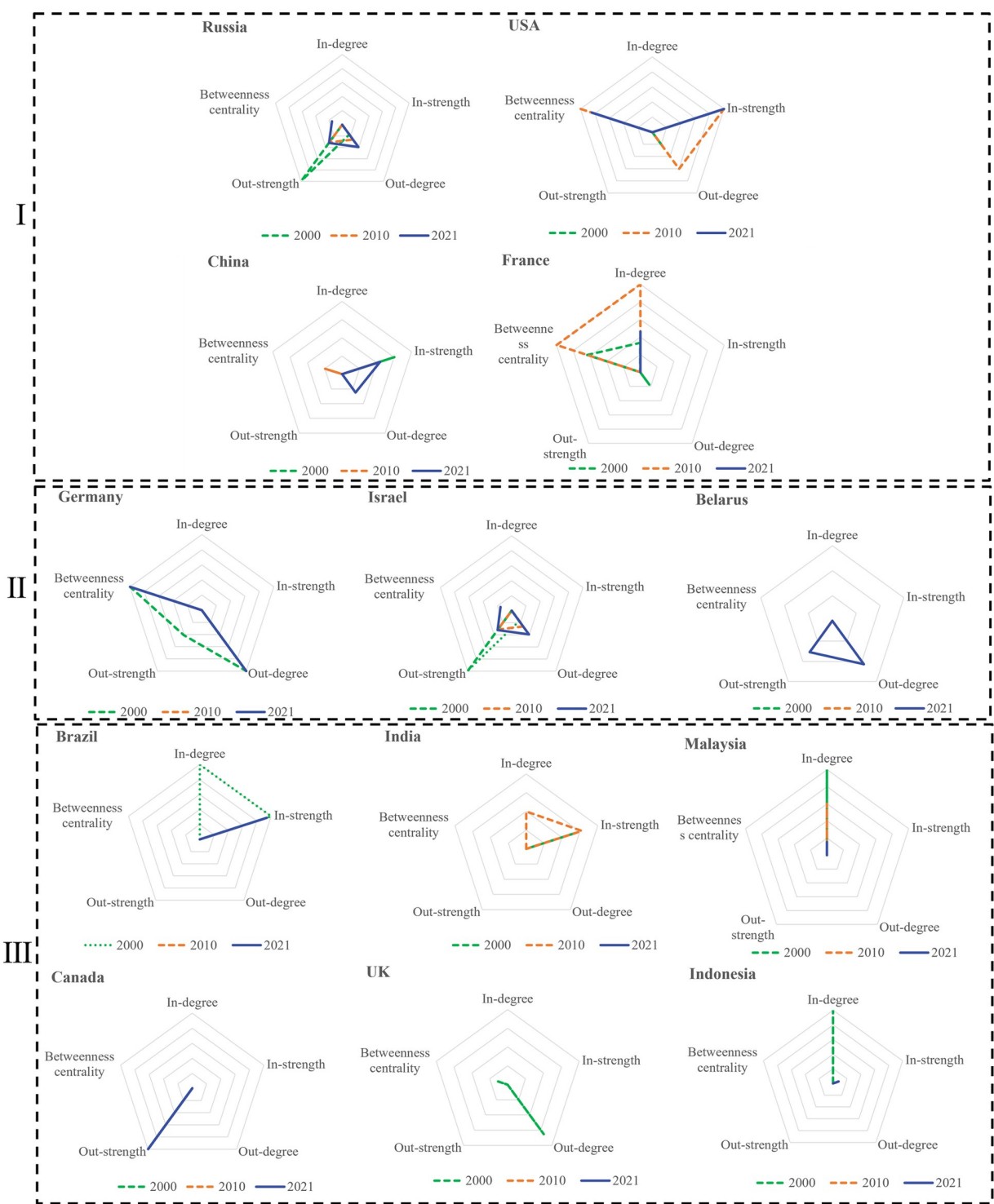

**Fig 5. The roles of 13 major countries in 2000,2010, and 2021.**

## 4. Conclusions and policy implications

This paper constructs a global potassium salt trade network from 2000 to 2021. We employ complex network theory to analyze the characteristics of the global potassium salt trade, the evolution of trade patterns, and the role of major trading countries in the network. The conclusions are as follows:

1. Overall, the potassium trade shows a trend of gradual prosperity, efficiency, and concentration. Besides, PTN is a scale-free network where minority countries have many trade relationships, and majority countries have few trade relationships. Furthermore, as an essential supplier and consumer country, China is moving closer to the center of the network, and its influence is growing.

2. The exporting countries of potassium remained stable, while imports changed significantly. The export of potassium salt is affected by the distribution of resources. The major exporting countries have remained stable in the past 20 years, mainly Canada, Russia, Belarus, Israel, Germany, and Jordan. The import market has changed significantly. The United States has gradually influenced China's import capacity, and the share of Canada's exports to China has shifted to the United States. With the United States showing significant import capacity in PTN, the center of the global potassium salt trade has shifted from Asia to America.

3. The evolution of the potassium trade community has characterized the fragmentation-regionalization-high concentration over time. In 2000, global potassium salt trade flows were fragmented; in 2010, the community evolution showed prominent geographical and spatial distribution characteristics, forming three major communities: North America, South America, and Europe; in 2021, the world's major potassium salt suppliers and consumers will gradually become a trading community, and trade relations will be highly concentrated. In addition, the top exporting country is the core country in the community, and different consuming countries join different trading communities according to their different trade preferences.

4. In PTN, the trade flow is unbalanced, and almost no country has shown significant capabilities. Exporting countries show strong export capabilities, and importing countries only show strong import capabilities. Only a few countries show certain import and export capabilities and intermediary capabilities. The role of major countries is divided into three levels based on trade relations, trade volume, and intermediary capabilities: the first tier is Russia, the United States, China, and France; the second tier is major suppliers such as Germany, Israel, and Belarus; the third tier is Brazil, India, Malaysia, Canada, the United Kingdom, and Indonesia, which show the relatively single capabilities.

Based on the above findings, we consider the current market environment and trade characteristics and make the following suggestions for the international trade of potassium salt: firstly, the exports of potassium salt are highly concentrated, it is recommended that importing countries seek more trading partners to reduce the risk of trade disruption; secondly, economies that export a large amount of potassium salt for a long time should re-examine their resource endowment status, rationalize their exploitation, and ensure the sustainability and long-term resource supply; thirdly, potassium salt trade presents a monopoly pattern. Major importing countries such as the United States, China, India, and Brazil are recommended to propose the initiative to establish an international agency to improve the cooperative relationship between potassium salt exporters and importers to maintain a stable, efficient, and sustainable market.

## Supporting information

**S1 Table. Node centrality of top 10 countries from 2000 to 2021.**
(DOCX)

## Acknowledgments

We would like to thank Mr. Wensong Zhang for his help in manuscript preparation and revision.

## Author Contributions

**Conceptualization:** Wensong Zhang.

**Data curation:** Bo Zhang.

**Formal analysis:** Bo Zhang.

**Funding acquisition:** Wensong Zhang.

**Investigation:** Bo Zhang.

**Methodology:** Wensong Zhang.

**Supervision:** Wensong Zhang.

**Writing – original draft:** Bo Zhang.

**Writing – review & editing:** Wensong Zhang.

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
