## [Decision Letter · Decision Letter 0]

31 Aug 2023

PONE-D-23-16699Uncovering the endogenous features of potassium salts' global transfer: a complex network perspectivePLOS ONE

Dear Dr. Zhang,

Thank you for submitting your manuscript to PLOS ONE. After careful consideration, we feel that it has merit but does not fully meet PLOS ONE’s publication criteria as it currently stands. Therefore, we invite you to submit a revised version of the manuscript that addresses the points raised during the review process.

We look forward to receiving your revised manuscript.

Kind regards,

Ayesha Afzal, PhD

Academic Editor

PLOS ONE

“This research is supported by grants from the National Natural Science Foundation of China (Grant No. 72074199 and No. 91846301).”

6. Please upload a copy of Figure 9, to which you refer in your text on page 15. If the figure is no longer to be included as part of the submission please remove all reference to it within the text.

7. We note that Figures 2 and 4 in your submission contain [map/satellite] images which may be copyrighted. All PLOS content is published under the Creative Commons Attribution License (CC BY 4.0), which means that the manuscript, images, and Supporting Information files will be freely available online, and any third party is permitted to access, download, copy, distribute, and use these materials in any way, even commercially, with proper attribution. For these reasons, we cannot publish previously copyrighted maps or satellite images created using proprietary data, such as Google software (Google Maps, Street View, and Earth). For more information, see our copyright guidelines: http://journals.plos.org/plosone/s/licenses-and-copyright.

a. You may seek permission from the original copyright holder of Figures 2 and 4 to publish the content specifically under the CC BY 4.0 license. 

8. We notice that your supplementary tables are included in the manuscript file. Please remove them and upload them with the file type 'Supporting Information'. Please ensure that each Supporting Information file has a legend listed in the manuscript after the references list.

Reviewers' comments:

Reviewer's Responses to Questions

**Comments to the Author**

1. Is the manuscript technically sound, and do the data support the conclusions?

Reviewer #1: Yes

2. Has the statistical analysis been performed appropriately and rigorously? 

Reviewer #1: N/A

3. Have the authors made all data underlying the findings in their manuscript fully available?

Reviewer #1: Yes

4. Is the manuscript presented in an intelligible fashion and written in standard English?

Reviewer #1: Yes

5. Review Comments to the Author

Reviewer #1: This is a very nice manuscript that presents an innovative analysis of the global trade in potassium salts, which are essential parts of the fertiliser trade and so sustain global food production. It uses international trade data from the year 2000, and presents an analysis of changes in the market since then. It cross-refers to United States Geological Survey publications that provide definitive information about the trade, from a geological perspective. The work has clearly been undertaken with great care.

As the manuscript states, with very few significant suppliers, global trade is very sensitive to disruption, with the war in Ukraine affecting the ability of Russia and Belarus to export potash. The analysis is timely, and emphasises the need of producing countries to address their ability to continue to export, and the need for importing countries to diversify the sources of their imports as far as possible.

Coming from an economic perspective, it is perhaps not surprising that the authors have not cited some of the recent (but pre-dating the war in Ukraine) publications on this topic, written from the perspective of mineral resources and agriculture. Recent reviews and analyses include:

Yakovleva, N., Chiwona. A. G., Manning, D. A. C. and Heidrich, O. (2021) Circular economy and six approaches to improve potassium life cycle for global crop production. Resources Policy, 74, 102426, https://doi.org/10.1016/j.resourpol.2021.102426

Ciceri, D, Manning, D. A. C. and Allanore, A. (2015) Historical and technical developments of potassium resources. Science of the Total Environment, 502, 590–601. https://doi.org/10.1016/j.scitotenv.2014.09.013

Manning, D. A. C. (2018) Innovation in resourcing geological materials as crop nutrients. Natural Resources Research, 27, 217-227. https://doi.org/10.1007/s11053-017-9347-2

Manning, D. A. C. (2010) Mineral sources of potassium for plant nutrition: a review. Agronomy for Sustainable Development, 30, 281-294. https://doi.org/10.1051/agro/2009023

The analysis that has been carried out is clearly explained, but I think that non-specialists may need more help in interpreting the text and the figures. For example, in Figure 3, maybe the major exporting countries could be named in bold or upper case text, so they are clearly visible (the arrows that cross the diagram are necessarily broad, so at first it is difficult to see the direction). Again, in Figure 4 the arrowheads are hard to see as they are at the end of the arrow; it would help if arrowheads were drawn also in the middle of the arrow, to emphasise the flow. Figure 4 could be clarified if the three categories were briefly summarised in the caption or indeed with additional text in the figure. I appreciate this is done in the main text. However, the groups are there described in terms of groups of countries, without explaining simply, ie in a very few words, why they are grouped as they are. This would help readers from other disciplinary backgrounds.

Figure 4 is especially interesting. Although some African countries are coloured, there are no arrows showing where their imports come from. Why is this? Could you comment on why Africa is not part of global trade in potassium salts? The following paper presents an early agronomic analysis of this:

Sheldrick W.F. and Lingard J. (2004). The use of nutrient audits to determine nutrient balances in Africa. Food Policy, 29, 61-98. https://doi.org/10.1016/j.foodpol.2004.01.004

An additional topic that could be considered in the Discussion is the possibility for disruption of the market by new products. The example I have in mind is polyhalite, and the Anglo American Woodsmith Mine (https://uk.angloamerican.com/the-woodsmith-project) that is designed to produce 20 million tonnes of polyhalite, an alternative to conventional potassium salts, for sale into the global fertilizer market. This will compete with conventional sources, and could have a major impact on trade patterns. However, although there is some trade in this material from an existing mine (ICL Boulby), it is very small. I believe, however, that China has large deposits of polyhalite that are being considered for exploitation, or are being mined at present. It would be good if the authors could comment on these new materials, although of course a similar analysis of their trade is not yet possible.

Minor comments relate mainly to the few occasions where correction to the excellent written English would improve the manuscript.

Page 1 Introduction first line: ‘are’ not ‘is’, as the verb refers to ‘fertilizers’

Introduction paragraph 3: here you should refer to some of the other papers that describe the global market in potassium, from different disciplinary perspectives.

Paragraph starting ‘Trade networks’ – do you mean ‘gold’ where the word ‘goal’ is written? Also make sure there is a space before the parenthesis when citing other papers.

Paragraph starting ‘In summary’ – line 2: ‘these’, not ‘there’.

Section 3.1.2: give the year for the citation of Yuxuan et al.

Please rewrite the sentence that follows this and describes ‘In international trade…’. As written it is difficult to understand.

Section 3.3 paragraph 2: please clearly separate the 3 groups of countries. At the moment, the second and third are described after a semicolon, making this a very long sentence. There should be a separate sentence for each group.

6. PLOS authors have the option to publish the peer review history of their article (what does this mean?). If published, this will include your full peer review and any attached files.

Reviewer #1: **Yes: **David Manning

---

## [Author Response · Author response to Decision Letter 0]

8 Nov 2023

Dear editor and reviewer:

Thank you for your comments concerning our manuscript entitled “Uncovering the endogenous features of the potassium salts’ global transfer: a complex network perspective”. Those comments are really valuable and helpful for revising and improving our paper. We have studied the comments carefully and revised our paper according to your advices. We responded to your comments point by point as follows:

Editor #1: Manuscript Number: PONE-D-23-16699

Title: Uncovering the endogenous features of the potassium salts’ global transfer: a complex network perspective

Journal: Plos One

[Answer] Thank you for your suggestion. We have revised the manuscript format according to the journal's requirements. Besides, we have uploaded the related files when resubmitting the manuscript.

Reviewer #1：Manuscript Number: PONE-D-23-16699

Title: Uncovering the endogenous features of the potassium salts’ global transfer: a complex network perspective

Journal: Plos One

Comment 1: Coming from an economic perspective, it is perhaps not surprising that the authors have not cited some of the recent (but pre-dating the war in Ukraine) publications on this topic, written from the perspective of mineral resources and agriculture. Recent reviews and analyses include:

[Answer] Thank you for your suggestion. We have carefully read these papers, which is very helpful for improving our manuscript. Since this manuscript focuses on the risk of trade networks, we did not cite all of them and selected two.

Comment 2: The analysis that has been carried out is clearly explained, but I think that non-specialists may need more help in interpreting the text and the figures. For example, in Figure 3, maybe the major exporting countries could be named in bold or upper case text, so they are clearly visible (the arrows that cross the diagram are necessarily broad, so at first it is difficult to see the direction). Again, in Figure 4 the arrowheads are hard to see as they are at the end of the arrow; it would help if arrowheads were drawn also in the middle of the arrow, to emphasise the flow. Figure 4 could be clarified if the three categories were briefly summarised in the caption or indeed with additional text in the figure. I appreciate this is done in the main text. However, the groups are there described in terms of groups of countries, without explaining simply, ie in a very few words, why they are grouped as they are. This would help readers from other disciplinary backgrounds.

[Answer] Thank you for your suggestion. We have adjusted Fig.3 and Fig.4 and made the font size of the major exporting countries in Fig.3 larger. Moreover, in Fig.4, we have removed the arrow of trade relations and determined the direction of trade by the color of the line. In addition, we explain how to determine the group in a few words. The revisions we made are as follows:

“As per Eq. (7), this paper quantifies the modularity of countries in the PTN and subsequently categorizes communities according to their modularity.”

Comment 3: Figure 4 is especially interesting. Although some African countries are coloured, there are no arrows showing where their imports come from. Why is this? Could you comment on why Africa is not part of global trade in potassium salts? 

[Answer] Thank you for your suggestion. This misunderstanding occurred because we did not explain it clearly. In Fig.4, the same color indicates the same community. Trade flows only include major trade countries. The figure will look very messy if we put all the trade flows. Therefore, we screen out trade relationships with relatively small trade volumes.

Comment 4: An additional topic that could be considered in the Discussion is the possibility for disruption of the market by new products. The example I have in mind is polyhalite, and the Anglo American Woodsmith Mine (https://uk.angloamerican.com/the-woodsmith-project) that is designed to produce 20 million tonnes of polyhalite, an alternative to conventional potassium salts, for sale into the global fertilizer market. This will compete with conventional sources, and could have a major impact on trade patterns. However, although there is some trade in this material from an existing mine (ICL Boulby), it is very small. I believe, however, that China has large deposits of polyhalite that are being considered for exploitation, or are being mined at present. It would be good if the authors could comment on these new materials, although of course a similar analysis of their trade is not yet possible.

[Answer] Thank you for this essential information. After our discussion, we prefer to focus on the trade issue of potassium salts. For polyhalite, this may be covered in future research. However, we have added a small discussion about polyhalite to arouse readers' thinking. The revisions we made are as follows:

“Notably, the discovery and development of potassium deposits in Canada have changed the global supply pattern. The UK's plans for large-scale polyhalite mining will provide an additional source of potassium, which may once again change the global supply pattern.”

Comment 5: Minor comments relate mainly to the few occasions where correction to the excellent written English would improve the manuscript.

[Answer] Thank you for your suggestion. These minor comments are essential for improving the manuscript. We revised the manuscript based on these comments.

In addition, we tried our best to improve the manuscript and made some changes in the manuscript. These changes will not influence the content and framework of the paper. And here we did not list the other changes.

We appreciate for Editors/Reviewers’ warm work earnestly, and hope that the correction will meet with approval.

Once again, thank you very much for your comments and suggestions.

---

## [Editor Report · Decision Letter 1]

16 Nov 2023

Uncovering the endogenous features of potassium salts' global transfer: a complex network perspective

PONE-D-23-16699R1

Dear Dr. Zhang,

We’re pleased to inform you that your manuscript has been judged scientifically suitable for publication and will be formally accepted for publication once it meets all outstanding technical requirements.

Kind regards,

Ayesha Afzal, PhD

Academic Editor

PLOS ONE

Additional Editor Comments (optional):

All of reviewer's comments have been adequately addressed and the manuscript is now suitable for publication. 
---

## [Editor Report · Acceptance letter]

21 Nov 2023

PONE-D-23-16699R1 

Uncovering the endogenous features of potassium salts’ global transfer: a complex network perspective 

Dear Dr. Zhang:

I'm pleased to inform you that your manuscript has been deemed suitable for publication in PLOS ONE. Congratulations! Your manuscript is now with our production department. 

Kind regards, 

on behalf of

Dr. Ayesha Afzal 

Academic Editor

PLOS ONE